# “Grafting-from” and “Grafting-to” Poly(N-isopropyl acrylamide) Functionalization of Glass for DNA Biosensors with Improved Properties

**DOI:** 10.3390/polym16202873

**Published:** 2024-10-11

**Authors:** Pauline Skigin, Perrine Robin, Alireza Kavand, Mounir Mensi, Sandrine Gerber-Lemaire

**Affiliations:** 1Group for Functionalized Biomaterials, Institute of Chemical Sciences and Engineering, Ecole Polytechnique Fédérale de Lausanne, EPFL SB ISIC SCI-SB-SG, Station 6, CH-1015 Lausanne, Switzerland; 2ISIC-XRDSAP, EPFL Valais-Wallis, Rue de l’Industrie 17, CH-1951 Sion, Switzerland; mounir.mensi@epfl.ch

**Keywords:** surface-based biosensor, poly(N-isopropyl acrylamide) (PNIPAM), thermoresponsive polymer, photoinduced electron/energy transfer reversible addition–fragmentation chain transfer polymerization (PET-RAFT), surface functionalization, antifouling, oligonucleotides

## Abstract

Surface-based biosensors have proven to be of particular interest in the monitoring of human pathogens by means of their distinct nucleic acid sequences. Genosensors rely on targeted gene/DNA probe hybridization at the surface of a physical transducer and have been exploited for their high specificity and physicochemical stability. Unfortunately, these sensing materials still face limitations impeding their use in current diagnostic techniques. Most of their shortcomings arise from their suboptimal surface properties, including low hybridization density, inadequate probe orientation, and biofouling. Herein, we describe and compare two functionalization methodologies to immobilize DNA probes on a glass substrate via a thermoresponsive polymer in order to produce genosensors with improved properties. The first methodology relies on the use of a silanization step, followed by PET-RAFT of NIPAM monomers on the coated surface, while the second relies on vinyl sulfone modifications of the substrate, to which the pre-synthetized PNIPAM was grafted to. The functionalized substrates were fully characterized by means of X-ray photoelectron spectroscopy for their surface atomic content, fluorescence assay for their DNA hybridization density, and water contact angle measurements for their thermoresponsive behavior. The antifouling properties were evaluated by fluorescence microscopy. Both immobilization methodologies hold the potential to be applied to the engineering of DNA biosensors with a variety of polymers and other metal oxide surfaces.

## 1. Introduction

Genosensors rapidly emerged as promising devices in the context of diagnostics, due to their favorable properties, including specificity, sensitivity, and fast outcomes [1,2]. They rely on the immobilization of DNA probes on a substrate that can interact with its targeted gene. This interaction can be further detected and interpreted by means of a signal transducer.

The performance of genosensors strongly depends on the chemistry used for the immobilization of the DNA probe, which dictates the resulting probe accessibility and grafting density [3]. These parameters are of notable importance to avoid a detrimental crowding effect, which hinders the access of the target RNA or DNA strand to the DNA probe for hybridization [3]. Increasing the distance between the probes and the surface has also shown to improve oligonucleotide orientation. In that regard, the use of short organic spacers [4,5,6,7] and polymeric layers [8] between the substrate and the DNA probes has been reported. In addition, the system must be designed to avoid non-specific surface adsorption, as it may lead to false-positive results. This is especially the case for the electrochemical, mass-based detection and surface-plasmon resonance detection of hybridization events [9,10,11,12].

We herein present a new functionalization strategy involving a temperature-sensitive polymeric layer spacer for the engineering of biosensors with improved properties. Coatings of thermoresponsive polymers have been extensively studied for diverse applications, such as in filtration membranes [13,14], drug delivery systems [15] or cell sheets. Among thermoresponsive polymers, poly(N-isopropylacrylamide) (PNIPAM) is the most studied material [16] in biomedical applications and was therefore selected for this study. Above a lower critical solution temperature (LCST) of around 32°C, PNIPAM undergoes a reversible phase transition from a soluble hydrated state to an insoluble dehydrated state. One of the reasons for the frequent use of this polymer for surface functionalization resides in its antifouling abilities, triggered by temperature changes [14,17,18]. On glass substrates, PNIPAM has mainly been grafted to control cell adhesion and detachment for cell culture applications [19,20] and for obtaining oriented calcium crystals, among others [21].

In this study, the conjugation of DNA probes to PNIPAM chains is therefore expected to provide the resulting sensing surfaces with antifouling properties, while allowing the immobilized DNA probes to hybridize with their targeted gene. Additionally, in 2015, Feng et al. [22] showed, via dissipative particle dynamics simulation, that conjugating DNA probes via a thermoresponsive polymer could also improve their orientation. We therefore believe our conjugation pathways to be of great interest for optimal DNA arrangement on 2D surfaces, to reach a high DNA hybridization density and, subsequently, a high sensitivity.

Many studies have reported the conjugation of DNA probes to polymer chains, but most of these approaches relied on the conjugation of the oligonucleotides to the backbone of the polymer [8]. To our knowledge, only one study [23] highlighted the conjugation of DNA probes to the end-group of a PNIPAM polymer on gold surfaces for biodetection purposes. However, this methodology relied on thiol–gold interactions, which cannot be transposed to other types of surfaces. In the present study, we describe and compare two covalent functionalization strategies to conjugate DNA probes via a thermoresponsive polymer chain on glass, a substrate frequently used for DNA biosensors. A first pathway (“grafting-from”) involves the anchoring of the chain transfer agent (CTA) through surface silanization, followed by the growth of a branched PNIPAM polymer using surface-initiated photoinduced electron transfer-reversible addition–fragmentation chain transfer polymerization (PET-RAFT). Subsequently, the thiocarbonylthio end-groups enable the post-polymerization functionalization of the polymers, leading to the conversion of these moieties into terminal carboxylic acids, followed by the covalent immobilization of ssDNA by amide bond formation. A second strategy, (“grafting-to”) relies on the modification of the glass surfaces with divinyl sulfone, followed by the grafting of thiolated-PNIPAM chains synthetized by RAFT polymerization and the attachment of the DNA probes via amide bond formation. Each step of the functionalization pathways was monitored via X-ray photoelectron spectroscopy (XPS). Finally, the properties of the surfaces in terms of DNA hybridization density, thermoresponsive behavior, and antifouling properties were evaluated. In this study, a Zika gene was chosen as the targeted gene; however, the designed methodology is independent from the probe sequence and could be applied to any other gene sequence.

## 2. Materials and Methods

***Preparation of S-Sil-CTA.*** Borosilicate slides were dusted off with an argon flow and immersed in glass tubes containing 1 mg·mL^−1^ **APTES-CTA** (synthesis procedure is detailed in Appendix A) in dry toluene. The slides were incubated for 30 min (25 °C, 750 rpm), then washed with toluene and acetonitrile before being dried with argon flow. The surfaces were then transferred to clean tubes and incubated for an additional hour (80 °C, 200 rpm). The slides were then directly used for the polymerization step.

***Preparation of S-Sil-PNIPAM-CTA.*** A stock solution containing NIPAM (550 mg), DTTMP (47 mg), and 1,3,5-trioxane (100 mg) was prepared in dry DMSO (10 mL). In each glass tube (9 cm height, 12 mm internal diameter, 15 mm outer diameter), 1.5 mL of the stock solution, a solution of Eosin Y (150 µL, 9 mg mL^−1^ in DMSO), and a solution of NEt_3_ (150 µL, 13 mg·mL^−1^ in dry DMSO) was added. **S-Sil-CTA** slides were immersed, and the glass tubes were sealed before being thoroughly degassed with argon for 10 min. Polymerization was carried out by illuminating the tubes under a 465–470 nm lamp for 5 h. The reaction was terminated by switching off the light source, and the slides were washed with DMSO, acetonitrile, and dichloromethane. The resulting **S-Sil-PNIPAM-CTA** slides were dried with argon flow and then stored under argon at −20 °C before being processed in the next step.

***Preparation of S-Sil-PNIPAM-SH.*** A stock solution of Na_2_S_2_O_4_ (16 mg) and propylamine (50 µL) was prepared in dry THF (10 mL). The **S-Sil-PNIPAM-CTA** slides were sealed in glass tubes, which were degassed with argon for 10 min before adding 2 mL of the stock solution per tube. After incubating for 4 h (25 °C, 750 rpm), the slides were washed with THF and dichloromethane and dried with a flux of argon. The resulting **S-Sil-PNIPAM-COOH** slides were stored under an inert atmosphere at −20°C before being processed in the next step.

***Preparation of S-Sil-PNIPAM-COOH.*** The **S-Sil-PNIPAM-SH** slides were sealed into glass tubes and degassed with argon for 10 min, before adding a solution of TCEP (2 mL, 1 mg·mL^−1^ in MilliQ water). After incubation for 3 h (25 °C, 750 rpm), the slides were washed with MilliQ water and 1,4-dioxane before being sealed and degassed with argon for 10 min in new test tubes. A solution of succinic anhydride (2 mL, 1 mg·mL^−1^ in 1,4-dioxane) was added, and the reactor was incubated overnight (25 °C, 750 rpm). The resulting **S-Sil-PNIPAM-COOH** slides were then washed with 1,4-dioxane and dichloromethane and dried with a flux of argon.

***Preparation of S-Sil-PNIPAM-DNA.*** A 0.1 M MES (2-(N-morpholino)ethanesulfonic acid) solution containing HOBt (60 mM) and EDC·HCl (50 mM) was prepared. In each tube, 2 mL of this solution was dispensed. The amino-modified ssDNA probe (10 mL of a 100 mM solution in MilliQ water) (sequence can be found in Appendix A) was added, and the slides were incubated for 24 h (25 °C, 750 rpm). The slides were then rinsed with MilliQ water and sealed in clean tubes to be washed with Tween 0.1% (twice) for 10 min (25 °C, 750 rpm). The **S-Sil-PNIPAM-DNA** functionalized slides were finally washed with MilliQ water and stored at 4°C.

***Preparation of S-Sulf.*** Borosilicate slides were treated for 20 min in a solution of KOH (50 mM) and hydrogen peroxide (7.5% v. of hydrogen peroxide solution) before being thoroughly rinsed with MilliQ water. The slides were immersed in 2 mL of a solution of divinyl sulfone (DVS, 200 mM) and triphenylphosphine (PPh3, 20 mM) in dry acetonitrile. The reactor was incubated overnight (60 °C, 750 rpm). The slides were then washed with MilliQ water, acetonitrile, and dichloromethane, before being dried with a flux of argon.

***Preparation of S-Sulf-PNIPAM-COOH.* COOH-PNIPAM-SH** (synthesis described in Appendix A) (1 mg mL^−1^) and TCEP·HCl (10 mM) were dissolved in HEPES buffer (2 mL, 0.1 M, pH 8.5). The **S-Sulf** slides were immersed in this solution, and the reactor was incubated overnight (25 °C, 750 rpm). The slides were then rinsed with MilliQ water and acetonitrile.

***Preparation of S-Sulf-PNIPAM-DNA.*** A 0.1 M MES solution containing HOBt (60 mM) and EDC·HCl (50 mM) was prepared. In each tube, 2 mL of this solution was dispensed. The amino-modified ssDNA probe (10 mL of a 100 mM solution in MilliQ water) was added and the slides were incubated for 24 h (25 °C, 750 rpm). The slides were then rinsed with MilliQ water and sealed in clean tubes to be washed with Tween 0.1% (twice) for 10 min (25 °C, 750 rpm). The **S-Sulf-PNIPAM-DNA** functionalized slides were finally washed with MilliQ water and stored at 4 °C.

## 3. Results

### 3.1. Design of the System

The two functionalization pathways designed to covalently immobilize oligonucleotides on glass surfaces via a thermoresponsive polymer are illustrated in Figure 1. The “grafting-from” pathway involved an initial silanization step on borosilicate slides with a derivative of (3-aminopropyl)triethoxysilane (APTES) to introduce the CTA (synthesis and characterization of the silanization reagent **APTES-CTA** are detailed in Appendix A). 2-(Dodecylthiocarbonothioylthio)-2-methylpropionic Acid (DTTMP) was chosen as the most suitable CTA for the acrylamide monomer [24]. In addition, inspired by several reports on the beneficial effects of post-silanization heating, we introduced a curing step at 80°C to produce **S-Sil-CTA** slides. Heat treatment was reported as an annealing step for silanized surfaces and as a promotor for the formation of a chemical bonding network between surface silanol groups and silane molecules [25,26]. Such protocol could therefore have a significant impact on the performance of the following polymerization and DNA conjugation steps. For instance, the immobilization of oligonucleotides on glass substrates functionalized with a bifunctional polymer brush was improved by elevating the curing temperature for the formation of initiator films based on 2-bromo-2-methyl-*N*-(3-triethoxysilylpropyl)-propionamide [27]. The next functionalization steps included polymer growth from the silanized surface through PET-RAFT induced by visible light illumination (465 nm) in the presence of Eosin Y as a photoinitiator and an NIPAM monomer. PET-RAFT was selected over conventional RAFT as it allows to carry the modification at room temperature [28] and is less sensitive to the presence of oxygen [29]. Then, the **S-Sil-PNIPAM-CTA** slides were subjected to aminolysis to release terminal thiols, which were further reacted in the presence of succinic anhydride. The resulting **S-Sil-PNIPAM-COOH** surface displayed reactive carboxylic groups for final conjugation to the amino-modified nucleotide via a peptide-coupling reaction.

The “grafting to” method was initiated with the catalytic oxa-Michael reaction of the surface silanol groups with divinyl sulfone, as reported by Cheng et al. [30]. In parallel, **PNIPAM-SH** was prepared via the RAFT polymerization of NIPAM, followed by end-chain functionalization. Then, the obtained polymer was reacted with the vinyl sulfone-modified surfaces in the presence of TCEP·HCl to avoid the oxidation of the thiol groups. Finally, amino-modified probes were conjugated to the carboxylic acids of the PNIPAM chains following the same procedure as for the “grafting from” pathway.

While many studies reported the conjugation of DNA probes to polymer brushes, these approaches mostly relied on the conjugation of the probes to the backbone of the polymer [8]. Herein, the designed conjugation strategies allow for the immobilization of DNA probes on the end-group of the polymer chain, which we expect to provide control over the probes’ orientation and sufficient lateral spacing.

### 3.2. “Grafting-from”: Synthetic Strategy Development in Solution and Surface Functionalization

#### 3.2.1. Photoinduced Radical Polymerization

Prior to the preparation of the sensing surface via the “grafting-from” method, all reactions involved in the surface functionalization strategy were studied in a solution to determine the optimal conditions for PET-RAFT polymerization. PNIPAM chains of around 30 monomers were chosen as a compromise to display a thermoresponsive behavior [16] and ensure a sufficient density of the covering layers for monitoring the evolution of surface atomic contents by XPS. Therefore, all the reactions were carried out with a [30]:[1]:[0.01]:[1] molar ratio (NIPAM/DTTMP/Eosin Y/TEA), under irradiation from blue LED light (λ_max_ = 465 nm, 130–140 lm).

To identify the suitable solvent, monomer conversion was monitored by ^1^H NMR spectroscopy in THF and DMSO (protocol and ^1^H-NMR spectra are provided in Appendix A). 1,3,5-trioxane was added to the reaction medium as an internal standard for NMR analysis to measure monomer conversion. ^1^H NMR spectra of the reaction medium were recorded at t = 0 and t = 3h, highlighting the higher conversion in DMSO (55%) than in THF (32%). DMSO was thus selected for further studies.

A kinetic study of the polymerization was performed in DMSO in order to determine the optimal reaction time. Samples were collected every hour from t = 0 to t = 5 h and analyzed by ^1^H NMR spectroscopy to measure monomer conversion at each time point (Figure 1a). The monomer disappeared at a steady rate during the first two hours, and the reaction decelerated between the second and third hour, until a 65% monomer conversion was reached after five hours (Figure 1b). The non-linear trend observed in the monomer conversion versus the reaction time plot is likely due to the combined effects of increasing viscosity in the polymerization solution and a decreasing monomer concentration. However, termination reactions were minimized, as evidenced by the linear fit in the plot of ln([M]_0_/[M]) against the reaction time, indicating an overall well-controlled polymerization (Figure 1c).

The collected samples were analyzed by gel permeation chromatography (GPC) and the growth of the polymer chains was validated by the number average molecular weight (M*_n, GPC_*) measurements (Figure 1d). The increase in M*_n,GPC_* in the first two samples was consistent with the observed fast monomer conversion. The polymer growth was then expectedly slower, until the molecular weight reached 4762 g mol^−1^ in the sample collected after a 5h reaction time.

#### 3.2.2. Post-Polymerization Modifications

The post-polymerization modifications required the derivatization of the end-group of the PNIPAM chain to introduce a carboxylic acid for further conjugation to amino-modified DNA probes (Figure 2). These steps were assessed on a batch of PNIPAM produced by conventional polymerization with AIBN as a thermal radical initiator, which enabled a more straightforward purification procedure and upscaling compared to photoinduced polymerization (see Appendix A). A [30]:[1]:[0.2] (NIPAM/DTTMP/AIBN) molar ratio was applied to obtain **PNIPAM-CTA** (experimental protocol and characterization are detailed in Appendix A), with a molar mass similar to the one obtained by photoinduced radical polymerization (M*_n, GPC_* = 5319 g mol^−1^).

Aminolysis in the presence of propylamine converted the RAFT agent into a thiol. The addition of sodium hydrosulfite as an antioxidant prevented the oxidative coupling of the thiol end-groups [31]. The resulting **PNIPAM-SH** polymer was purified by precipitation in diethyl ether. Reductive treatment with TCEP followed by condensation with succinic anhydride afforded **PNIPAM-COOH** in a 30% yield after purification by dialysis. While the isolated **PNIPAM-SH** polymer did not seem to dimerize (M_n_ = 4839 g mol^−1^), the addition of TCEP was necessary to ensure successful condensation with succinic anhydride. The reaction sequence was monitored by ^1^H NMR analysis of intermediate and final polymers (Appendix A). In addition, 2D DOSY NMR analysis (Appendix A) gave evidence for the covalent conjugation of the succinic spacer to the polymer chain. **PNIPAM-SH** and **PNIPAM-COOH** were analyzed by GPC (see Appendix A), and the measured molecular weight values were consistent with the functionalization steps.

#### 3.2.3. Surface Functionalization

Silanization of borosilicate slides in the presence of **APTES-CTA** (30 min, 25 °C, dry toluene), followed by thermal treatment (80 °C, 1.5 h), afforded **S-Sil-CTA** slides (Figure 1, “grating from” pathway), which were analyzed by XPS. The relative atomic concentrations are displayed in Table 1. The increase in C 1s and N 1s content and the high-resolution XPS spectra of the atomic signals were consistent with successful surface silanization (Figure 2a,b).

Then, the conditions developed in a solution were implemented for the polymer growth and post-polymerization on CTA-modified surfaces to afford **S-Sil-PNIPAM-COOH** slides (Figure 1). Based on the molecular weight of the polymers collected after PET-RAFT in solution, we estimated the M_n_ of the PNIPAM polymers grown on **S-Sil-CTA** slides to reach values around 4000 g mol^−1^ [32,33]. The 5′-amino-modified ZAS DNA probe (sequence detailed in Appendix A) was conjugated to the end-carboxylic groups following our previously reported method [34]. Intermediate and final slides were analyzed by XPS, monitoring the evolution of the C 1s and N 1s content through the calculation of the C/Si and N/Si ratios (Table 1). In addition, covalent coupling to the DNA sequence was evidenced by the increase in the N 1s signal (Table 1) and the appearance of a P 2p signal on the **S-Sil-PNIPAM-DNA** slides (Figure 2c).

### 3.3. “Grafting-to”: Polymer Synthesis and Surface Functionalization

#### Polymer Synthesis and Surface Functionalization

The **PNIPAM-SH** polymer needed for the implementation of the “grafting-to” pathway was prepared according to the protocol described in Figure 2. Then, the borosilicate slides were treated with divinyl sulfone, in the presence of PPh_3_ [23], and the resulting **S-Sulf** slides were analyzed by XPS. The relative atomic concentrations are given in Table 2. The increase in C 1s content and the apparition of a S 2p signal (Figure 3) were consistent with successful surface modification with divinyl sulfone.

The subsequent thiol Michael addition of **PNIPAM-SH** to the surface-conjugated vinyl sulfone groups was investigated in different reaction conditions, including (i) polymer pre-treatment with TCEP, followed by incubation in PBS buffer (pH 7), (ii) polymer pre-treatment with TCEP, followed by incubation in HEPES buffer (pH 8.5), and (iii) incubation with the polymer and TCEP in HEPES buffer (pH 8.5). The last condition led to the highest DNA hybridization density and was therefore selected for the rest of the study. The 5′-amino-modified DNA probe was then conjugated to the end-carboxylic groups of the polymer through amide coupling. Intermediate and final slides were analyzed by XPS, monitoring the evolution of the C 1s and N 1s content through the calculation of C/Si and N/Si ratios (Table 2). In addition, the appearance of a P 2p signal in the **S-Sulf-PNIPAM-DNA** slides indicated the successful immobilization of DNA probes (Figure 4).

Additional characterization techniques were investigated to assess the density and thickness of the polymer on the surfaces, post functionalization. AFM measurements were performed after scratching the surface to remove the polymer layer to measure the step height across the step formed. Unfortunately, such measurements did not provide reliable results due to the suboptimal smoothness of the surface on the glass substrates. Additionally, ellipsometry was attempted on the DNA-conjugated surfaces. Due to the transparency of the materials, such measurements could not be interpreted to evaluate the polymer thickness or density on the surfaces. This study therefore highlights the issues faced when characterizing functionalized glass surfaces, unless the grafting density is sufficiently high to be characterized as polymer brushes [19,35]. As glass substrates are frequently used in the preparation of DNA-conjugated surfaces [36], more sensitive techniques would be of great interest to evaluate coating thickness and lower-density grafted quantities and/or smaller molecules.

### 3.4. Surface Properties of Functionalized Surfaces

#### 3.4.1. DNA Hybridization Density

The capability of the DNA-conjugated surfaces to hybridize with the targeted gene was evaluated through a fluorescence assay developed in our previous study (Figure 3, detailed experimental protocol in Appendix A). Following the incubation of **S-Sil-PNIPAM-DNA** or **S-Sulf-PNIPAM-DNA** with a Cy3-tagged targeted gene sequence and the subsequent thermal denaturation of the duplexes, the fluorescence quantified in the supernatant indicated hybridization density values of 3.09 ± 0.98 pmol·cm^−2^ (n = 4) and 1.49 ± 0.10 pmol·cm^−2^ (n = 3), respectively. The hybridization density of **S-Sil-PNIPAM-DNA** stands among the highest values reported so far in the literature for surface-based DNA biosensors.

The hybridization density is one of the most critical parameters affecting the performance of a biosensor, especially in terms of sensitivity [2]. Previous studies on glass substrates covered with polymeric layers, such as poly(PEGDA-co-GMA) [37], divinyl sulfone [38], and cyclic olefin copolymer [39], reported hybridization density values below 0.5 pmol cm^−2^. On PMMA substrates, Miyahara et al. [40] described the use of methyl methacrylate and 2-hydroxyethyl methacrylate dropcast co-polymers to reach values from 1.0 ± 0.8 to 5.4 ± 1 pmol cm^−2^. The implementation of a PNIPAM-based polymeric layer on a glass substrate, as herein described, resulted in a high hybridization density, depending on the chemical pathway followed. Noteworthily, the methodology should be suitable for any type of amino-modified ssDNA sequences and a variety of substrates, especially metal oxides.

#### 3.4.2. Thermoresponsive Behavior

The thermoresponsive properties of the functionalized surfaces were verified by measuring the water contact angle (WCA) of both **S-Sil-PNIPAM-DNA** and **S-Sulf-PNIPAM-DNA** below and above the LCST of PNIPAM (around 30°C in solution in pure water, Appendix A). Pictures of water droplets deposited on the surface of borosilicate (control) and DNA-conjugated slides are shown in Figure 5. At 20 °C, the WCAs on the surface of **S-Sil-PNIPAM-DNA** and **S-Sulf-PNIPAM-DNA** were, respectively, 45.3 ± 2.5 and 24.2 ± 0.5 degrees (n = 3 independent measurements). The higher WCA value on **S-Sil-PNIPAM-DNA** can be explained by the highly hydrophobic initial alkyl silane layer. Upon heating to 50 °C, the WCA values increased to 57.2 ± 0.1 and 66.6 ± 3.5 degrees for **S-Sil-PNIPAM-DNA** and **S-Sulf-PNIPAM-DNA**, respectively. The WCA shift toward higher values when transitioning from 20 to 50 °C confirmed the thermoresponsive behavior of the functionalized surfaces resulting from the presence of a PNIPAM covering layer as the borosilicate slide did not show any dependance of the WCA values on the temperature of the measurement. The WCA was measured at 5 °C intervals between 20 and 50 °C for **S-Sil-PNIPAM-DNA** and **S-Sulf-PNIPAM-DNA** (Appendix A). A clear shift was observed between 30 and 40 °C for **S-Sil-PNIPAM-DNA**, while the WCA values increased regularly for **S-Sulf-PNIPAM-DNA** between 20 and 40 °C, where they reached a plateau.

#### 3.4.3. Antifouling Properties

The antifouling behavior of the surfaces was evaluated by fluorescence microscopy, following a procedure inspired by Parviz et al. [41]. A solution of bovine serum albumin (BSA) labeled with Alexa Fluor dye 488 (BSA-AF488) was deposited on the surfaces, which were incubated at 22 °C and 42 °C for one hour. The slides were then washed with PBS to remove the weakly bound BSA, and the surfaces were imaged via fluorescence microscopy, as shown in Figure 6.

Both **S-Sil-PNIPAM-DNA** and **S-Sulf-PNIPAM-DNA** showed a decrease in fluorescence levels upon incubation at 42 °C, compared to 22 °C (Figure 6b). This is consistent with the fact that BSA adsorbs in larger proportions on hydrophilic surfaces than hydrophobic surfaces [42].

The substrates prepared via the “grafting-to” pathway showed lower protein fouling at both temperatures. These results indicate the capacity of a PNIPAM layer between the glass substrate and the DNA probes to reduce protein absorption above the LCST of the immobilized polymer.

Overall, both the “grafting-from” and “grafting-to” methodologies led to the engineering of DNA-conjugated glass surfaces with high DNA hybridization density, thermoresponsive properties, and antifouling behavior. The “grafting-to” pathway is more straightforward, consisting of only three steps starting from borosilicate substrates, which can be performed in three days. On the contrary, the “grafting-from” pathway requires seven steps and necessitates five processing days. **S-Sil-PNIPAM-DNA** samples were characterized by a higher hybridization density, while **S-Sulf-PNIPAM-DNA** displayed enhanced antifouling properties. Also, the “grafting-to” methodology allows for more precise control over the polymer’s characteristics before its immobilization on coated surfaces. Nonetheless, the “grafting-from” pathway should be selected in applications requiring higher hybridization densities. Both strategies were designed to be adaptable to a wide variety of substrates, including metal oxides and potentially plastics for the “grafting-to” process.

As previously reported in a computational study, conjugating DNA probes via thermoresponsive polymer chains is also expected to improve DNA orientation upon temperature variation [22]. This parameter is crucial to achieving an optimal sensing performance, as poorly oriented oligonucleotides are less prone to hybridization with their targeted genes. While no characterization technique has yet been identified to verify this parameter on glass surfaces, further studies could compare the sensing capacities of these surfaces to others prepared with non-thermoresponsive polymers, to verify the positive impact of employing such polymers.

## 4. Conclusions

In this study, we presented the first methodology for the conjugation of DNA probes on glass substrates via the end-group of a thermoresponsive polymer chain, here grown by RAFT polymerization. While previous strategies focused on the immobilization of nucleic acid sequences on the polymer backbone, we took advantage of end-group modifications to ensure covalent immobilization and proper DNA orientation. While no technique was yet identified to verify the probes’ orientation, the surfaces resulted in remarkably high hybridization density values (3.09 ± 0.98 pmol·cm^−2^ for **S-Sil-PNIPAM -DNA**), which already suggests the vertical orientation of the oligonucleotide sequences. Both the “grafting-from” and “grafting-to” pathways were monitored by the XPS analysis of intermediate and final conjugated slides to track the evolution of the atomic contents characteristic to the polymeric chains (C 1s, N 1s) and the appearance of specific signals accounting for the oligonucleotide sequence (P 2p). The introduction of a PNIPAM layer was also expected to lead to thermoresponsive behavior and antifouling properties. The thermoresponsive properties of the PNIPAM spacing polymer were maintained upon immobilization and post functionalization, as illustrated by the WCA measurements. Also, the DNA-conjugated surfaces displayed antifouling properties when exposed to temperatures above the LCST of the polymer. However, this study was conducted with only one type of protein, and the impact of the antifouling properties on the detection abilities of the DNA-conjugated surfaces has not yet been verified.

Further investigations should therefore be pursued to study the DNA orientation resulting from covalent conjugation to PNIPAM layers and evaluate the advantages of thermoresponsive DNA biosensors for the detection of analytes in complex biological samples. As the designed functionalization pathways do not depend on the nature of the selected monomers, the methodology could be extended to a variety of other polymers.

## Data Availability

The datasets generated during the current study are available on Zenodo at https://doi.org/10.5281/zenodo.13772898.

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
