# Peer review of "“Grafting-from” and “Grafting-to” Poly(N-isopropyl acrylamide) Functionalization of Glass for DNA Biosensors with Improved Properties"

_polymers, 2024, doi:10.3390/polym16202873_

Round 1

Reviewer 1 Report

Comments and Suggestions for Authors

Pauline Skigin et al. describe and compare two functionalization methodologies to immobilize DNA probes on a glass substrate via a thermoresponsive polymer, in order to produce genosensors with improved properties. This is a very interesting concept and after minor revision could be considered for publication in polymers journal.

1. What are the advantages of the first method over the second?

2. What is the improvement of the sensor surface grafting method proposed in this manuscript over previously reported literature?

3. How long does it take to complete a surface graft like this?

4. How to detect the surface graft density of sensor DNA?

Author Response

• Comment 1. What are the advantages of the first method over the second?

Response: The first method (“grafting from”) can be more easily implemented on different types of surfaces, such as plastics for example, while the second one can be mainly used for metal oxides surfaces. It also leads to higher DNA hybridization density. The second method (“grafting to”), however, is more straightforward and requires a reduced number of synthetic steps. The paragraph at the end of section 3 has been further detailed to compare both methods.

Comment 2. What is the improvement of the sensor surface grafting method proposed in this manuscript over previously reported literature?

Response: The main advantage of these conjugation pathways resides in the fact that, contrary to most reported studies, the DNA probes are conjugated via the end group of the polymer chains, which is expected to improve the spatial configuration of the DNA probes on the surface. Also, a point was added at the end of section 3 to highlight the fact that these coating pathways, beside leading to surfaces with antifouling properties, are expected to improve the DNA orientation as shown by Feng et al. (Sci Rep 2015, 5, 14415, doi:10.1038/srep14415).

• Comment 3. How long does it take to complete a surface graft like this?

Response: These indications were added at the end of section 3.

• Comment 4. How to detect the surface graft density of sensor DNA?

Response: A paragraph was added to discuss about the characterization techniques to evaluate the grafting density/thickness of the polymer. In the frame of this study, ellipsometry and AFM measurements were conducted. Unfortunately, such techniques are not suitable for borosilicate substrates (unlike silicon for example). This paper therefore highlights that characterization techniques must be specifically further developed for glass substrates, which are frequently encountered for DNA biosensors preparation.

Reviewer 2 Report

Comments and Suggestions for Authors

This paper presents timely and interesting research on the fabrication and properties of PNIPAM-DNA conjugates. The authors developed two distinct methodologies for functionalizing glass surfaces with grafted PNIPAM brushes, followed by conjugation with DNA. While the paper is promising, it requires major revisions before it can be accepted for publication. Several key points need clarification.

First, the motivation for the study should be explained in more detail. The current discussion focuses mainly on the antifouling properties of DNA biosensors, but this argument appears weak given that only low-fouling properties were demonstrated in the study. In contrast, the use of thermoresponsive polymers to improve DNA orientation is a much stronger aspect, but it needs to be elaborated more thoroughly in the manuscript.

The acronym "MES" needs to be defined for clarity.

A critical point that requires attention is the amount of PNIPAM grafted to the surface. Information regarding the grafting densities and thickness of the brush coatings would significantly strengthen the paper and should be included.

It would also be beneficial to measure the LCST of the coatings, as LCST values on surfaces often differ from those in solution. In the supporting information shown in Fig. 6S, the authors claim, "Both polymers displayed similar LCST values in all media." However, the data show clearly different LCSTs in various media. This discrepancy must be corrected.

Lastly, I suggest referencing similar studies that have investigated PNIPAM-based systems. For example, the paper available at https://doi.org/10.1021/acsbiomaterials.3c00917 would be a valuable addition to the reference list.

Comments on the Quality of English Language

Minor editing of English language required.

Author Response

Comment 1. First, the motivation for the study should be explained in more detail. The current discussion focuses mainly on the antifouling properties of DNA biosensors, but this argument appears weak given that only low-fouling properties were demonstrated in the study. In contrast, the use of thermoresponsive polymers to improve DNA orientation is a much stronger aspect, but it needs to be elaborated more thoroughly in the manuscript.

Response: Unfortunately, there is currently no technique available to study the DNA orientation on this type of surfaces and with DNA probes of this length. Here, we based our design on a computational study (Feng et al. Sci Rep 2015, 5, 14415, doi:10.1038/srep14415) which showed that immobilizing the DNA probes at the end group of polymer chains would improve their orientation. Our study aimed at showing i) the feasibility of such coating on glass surface, and ii) the possibility to obtain sensing surfaces with a high DNA hybridization capacity with antifouling properties. The introduction and conclusions were modified to more clearly define the claims of this study and its limitations, and a paragraph has been added at the end of section 3 to discuss more in detail the orientation of the probes. It is also to be noted that the hybridization densities achieved on the surface (especially Surf-Sil-PNIPAM-DNA) are among the highest reported in the literature, which is critical for the sensitivity performance of the sensing surfaces, and already tend to indicate that the probe orientation was favorable.

•  Comment 2. The acronym "MES" needs to be defined for clarity.

Response: The acronym MES was clarified in the text.

Comment 3. A critical point that requires attention is the amount of PNIPAM grafted to the surface. Information regarding the grafting densities and thickness of the brush coatings would significantly strengthen the paper and should be included.

Response: A paragraph was added to discuss about the characterization techniques to evaluate the grafting density/thickness of the polymer. In the frame of this study, ellipsometry and AFM measurements were conducted. Unfortunately, such techniques are not suitable for borosilicate substrates (unlike silicon for example). This paper therefore highlights that characterization techniques must be specifically further developed for glass substrates, which are frequently encountered for DNA biosensors preparation.

• Comment 4. It would also be beneficial to measure the LCST of the coatings, as LCST values on surfaces often differ from those in solution. In the supporting information shown in Fig. 6S, the authors claim, "Both polymers displayed similar LCST values in all media." However, the data show clearly different LCSTs in various media. This discrepancy must be corrected.

Response: The sentence “Both polymers displayed similar LCST values in all media” was clarified; the buffer indeed has a drastic impact on the thermoresponsive behavior, but both polymers studied had in each of these different buffers, similar LCST. To improve clarity, we updated the Supporting Information and only displayed the measurements on the polymer PNIPAM-COOH, which was grafted to the coated surfaces. The data related to the LCST values on surfaces are disclosed in Figure S8, showing water contact angle values as a function of temperature. For S-Sil-PNIPAM-DNA, a LCST comprised between 30 and 40°C was observed, which is in accordance with the values measured in solution. For S-Sulf-PNIPAM-DNA, a steady increase was measured from 20°C to 40°C where a plateau value was reached.

Comment 5. Lastly, I suggest referencing similar studies that have investigated PNIPAM-based systems. For example, the paper available at https://doi.org/10.1021/acsbiomaterials.3c00917 would be a valuable addition to the reference list.

Response: This study and other references were added in the Introduction.

Comment 6. Minor editing of English language required.

Response: Language errors were corrected in the manuscript.

Reviewer 3 Report

Comments and Suggestions for Authors

The manuscript developed PNIPAM functionalization for potentially developing DNA-based biosensors. It demonstrates experimental findings to discuss how the PNIPAM is utilized for grafting the DNA sequences. While the authors fully demonstrated the final biosensor application, the study in the manuscript can still potentially interest potential readers. 

While the manuscript covers an exciting topic, several concerns arise regarding its novelty and depth of experimental analysis. The authors reviewed a few existing literature articles but did not offer sufficient new insights or groundbreaking findings to make this work highly novel. I suggest the authors elaborate the introduction and conclusion section to clearly state their work's limitations and future work to be done.

Comments on the Quality of English Language

No critical errors were found in the manuscript.

Author Response

• Comment 1. While the manuscript covers an exciting topic, several concerns arise regarding its novelty and depth of experimental analysis. The authors reviewed a few existing literature articles but did not offer sufficient new insights or groundbreaking findings to make this work highly novel. I suggest the authors elaborate the introduction and conclusion section to clearly state their work's limitations and future work to be done.

Response: In this paper, we provide evidence for i) the feasibility (via two different pathways) to covalently conjugate DNA probes via the end group of a thermoresponsive polymer chains on glass surfaces. One of our pathways could be also easily transposed to many other types of surfaces; and ii) the high DNA hybridization capacity achieved, especially for the “grafting from” method (among the highest reported so far in the literature) combined with antifouling properties of the resulting surfaces, resulting from the polymer layer. In addition, from a previous reported computational study (Feng et al. Sci Rep 2015, 5, 14415, doi:10.1038/srep14415), we expect the engineered surfaces to display improved probe orientation. Further studies should be conducted to experimentally validate these computational predictions. The introduction and conclusion sections were therefore elaborated to better state the claims of the reported study, and the limitations of this work. A paragraph was specifically added at the end of section 3 to discuss about the DNA orientation parameter.

Round 2

Reviewer 2 Report

Comments and Suggestions for Authors

The authors have completely answered all my issues; the paper can be accepted in its present state.